

# Delineating the distribution of mineral and peat soils at the landscape scale in northern boreal regions

Anneli M. Ågren[1], Eliza Maher Hasselquist[1], Johan Stendahl[2], Mats B. Nilsson[1], Siddhartho S. Paul[1].

[1]Department of Forest Ecology and Management, Swedish University of Agricultural Science, Umeå, SE-90187, Sweden
[2]Department of Soil and Environment, Swedish University of Agricultural Science, Uppsala, SE-756 51, Sweden

*Correspondence to*: Anneli M. Ågren (anneli.agren@slu.se)

**Abstract.** A critical tool to succeed in sustainable spatial planning is accurate and detailed maps. To meet the sustainable development goals and enable sustainable management and protection of peatlands, there is a strong need for improving the mapping of peatlands. Here we present a novel approach to identify peat soils based on a high-resolution digital soil moisture map that was produced by combining airborne laser scanning-derived terrain indices and machine learning to model soil moisture at 2 m spatial resolution across the Swedish landscape with high accuracy (Kappa = 0.69, MCC = 0.68). As soil moisture is a key factor in peat formation, we fitted an empirical relationship between the thickness of the organic layer (measured at 5 479 soil plots across the country) and the continuous SLU soil moisture map ($R^2$ = 0.66, p < 0.001). We generated categorical maps of peat occurrence using three different definitions of peat (30, 40 and 50 cm thickness of the organic layer) and a continuous map of organic layer thickness. The predicted peat maps had a higher overall quality (MCC = 0.69-0.73) compared to traditional quaternary deposits maps (MCC = 0.65) and topographical maps (MCC = 0.61) and captured the peatlands with a recall of ca 80% compared to 50-70% on the traditional maps. The predicted peat maps identified more peatland area than previous maps, and the areal coverage estimates fell within the same order as upscaling estimates from national field surveys. Our method was able to identify smaller peatlands resulting in more accurate maps of peat soils, which was not restricted to only large peatlands visible from airplanes - the historical approach of mapping. Most importantly we also provided a continuous map of the organic layer, which ranged 6-95 cm organic layer thickness, with an $R^2$ of 0.67 and RMSE of 19 cm. The continuous map exhibits a smooth transition of organic layers from mineral soil to peat soils and likely provides a more natural representation of the distribution of soils. The continuous map also provides an intuitive uncertainty estimate in the delineation of peat soils, critically useful for sustainable spatial planning, e.g. green-house gas or biodiversity inventories and landscape ecological research.

## 1. Introduction

Soil, i.e. the pedosphere, provides a suite of unique and essential ecosystem services globally (Smith et al., 2021), such as food (Silver et al., 2021) and forest production (Laamrani et al., 2014), conservation of water resources (Cheng et al., 2021), modulation of extreme events (Saco et al., 2021), and regulation of the global carbon cycle (Scharlemann et al., 2014). The characteristics of the pedosphere depend on multiple soil forming factors, i.e.



parent material, climate, organisms, topography, and time (Jenny, 1941). In the northern boreal regions, the parent material of soil is mostly composed of quaternary deposits that were formed during several cycles of glaciation and deglaciation (Imbrie et al., 1993). Therefore, the surficial sediments in these regions are mainly characterized by unsorted deposits from preceding stadial and interstadial periods (Hirvas et al., 1988; Olsen et al., 2013). Sorted

sediments were also deposited in glaciofluvial deposits such as eskers, as well as marine, and lacustrine environments during and after the glacial melting (Stroeven et al., 2016). Furthermore, the humid climate of the northern boreal regions favors anoxic soil conditions that supports widespread mire formation and peat deposits (Ivanov, 1981; Rydin and Jeglum, 2013). Altogether, the boreal ecosystem contains one of the largest terrestrial carbon storages of the world (Beaulne et al., 2021; Loisel et al., 2014), which makes it significantly important for

the earth system, especially under the current global warming and climate change (Astrup et al., 2018). The boreal biome stores about 272 (±23) Pg of C and 60% of this carbon is found in soil organic matter (Pan et al., 2011).

The soil moisture regime, which is also a strong regulator of soil organic matter (SOM) dynamics, is a critical factor for ecosystem functioning and management in the boreal regions (Ivanov, 1981; Sewell et al., 2020). Soil

moisture and SOM feedback has been clearly documented, for example, in central and northern Sweden which comprises a key boreal forest region (Hounkpatin et al., 2021). In Swedish boreal podzols, dry sites have an average soil organic carbon (SOC) stock of 6.7 kg C $m^{-2}$ while mesic-moist sites had 9.7 kg C $m^{-2}$ in the mineral horizons and 2.0 to 4.4 kg C $m^{-2}$ alone in the organic horizon (Olsson et al., 2009). Moreover, SOC stock in peatlands of this boreal region is even higher, ranging from 22.6 to 72.0 kg C $m^{-2}$ (Beaulne et al., 2021; Loisel et

al., 2014). This clear relationship between SOM and soil moisture regime can provide an opportunity for mapping the distribution of peat soils in the boreal ecosystem. Soil maps detailing the distribution of mineral and peat soils across the boreal regions could support better the sustainable management and ecological restoration activities in these regions. Unfortunately, availability of soil maps are substantially limited for most regions and existing maps are mostly based on the technology and data from the 1900's – usually a result of manual interpretation of

topographical maps, aerial photos and field investigations (Olsson, 1999).

Recently, high-resolution topographic data from airborne laser scanning (ALS) has provided a new avenue for producing highly accurate maps of soil and site conditions at local to regional scales (Behrens et al., 2018; Latifovic et al., 2018; O'neil et al., 2020; Pouliot et al., 2019; Prince et al., 2020). In Sweden, for instance, ALS data at 2 m

spatial resolution were combined with machine learning (Lidberg et al., 2020) to map soil moisture condition in a recent study that exhibited both categorical soil wetness classes and continuous moisture variation at the national scale (Ågren et al., 2021). These maps provide new opportunities to explore the relationship between soil moisture regime and thickness of the organic layer, which in turn, can be used to map the horizontal distribution of peat and mineral soils at landscape scale. In this study, we used Sweden as a test area for mapping the distribution of peat

soils specifically based on soil moisture information, where we combined data from the National Forest Inventory (NFI) (Fridman et al., 2014), the Swedish Forest Soil Inventory (Stendahl et al., 2017), and the nation-wide soil moisture map (Ågren et al., 2021).

This study focuses on the division between peat and mineral soils based on soil moisture condition. Here, we

define peat soils and peatlands according to the definition provided by Rydin and Jeglum (2013). *"Peat is the*



*remains of plant and animal constituents accumulating under more or less water-saturated conditions owing to incomplete decomposition. It is the result of anoxic conditions, low decomposability of the plant material, and other complex causes. Peat is organic material that has formed in place, i.e. as sedentary material, in contrast to aquatic sedimentary deposits. Quite different plant materials may be involved in the process of peat formation, for*

*instance, woody parts, leaves, rhizomes, roots and bryophytes (notably Sphagnum peat mosses)"…" Peatland generally refers to peat-covered terrain while a minimum depth of organic layer is required for a site to be classified as peatland".* For technical or practical reasons, a minimum organic layer depth is commonly used to define peatlands. However, such technical depth-based definition of peatlands incorporates strong biases in aerial estimates of peatland across large regions. In addition, there is inconsistency - nationally and internationally -

about the minimum organic layer depth required for an area to be classified as a peatland. For example, the Geological Survey of Sweden sets a threshold of 50 cm organic layer depth for peatland. The same threshold is also used in Scotland (Burton, 1996) while an organic layer depth of 40 cm is recognized for peatland in Canada, England, and Wales (Burton, 1996; Cruickshank and Tomlinson, 1990; Zoltai et al., 1975). This 40 cm limit also follows the definition of "Histosols" (i.e. soils consisting of dominantly organic materials) according to the World

Reference Base for Soil Resources (WRB, 2015). Moreover, an organic layer thickness of 30 cm is used for defining peatland by the Swedish Forest Soil Inventory and NFI, Geological Survey of Finland, International Mire Conservation Group, and International Peat Society (Joosten and Clarke, 2002; Lappalainen and Hänninen, 1993). A shallower threshold of organic layer thickness to define a peatland will include more of the mineral-soil wetlands that often have a substantial content of organic matter within their surface layers. But the organic matter in these

mineral-soil wetlands has not had a high enough accumulation rate or has not had enough time for thicker peat formation. Such soils in the boreal region may include the soil type "peaty mor" and form landscape features such as "cryptic wetlands," which are usually elongated small areas with saturated soils commonly found in the bottom of small valleys, and riparian peat (Creed et al., 2003; Ploum et al., 2018; Kuglerova et al., 2014a). Therefore, to be useful for different practitioner groups and the scientific community, the map of peat soil distribution needs to

incorporate multiple definitions of peatland based on the thickness of organic layer.

In this study, we developed an approach for accurate mapping of peat soil distribution based on the relationship between soil moisture variation and organic layer thickness using Sweden as a test case for a peat-rich northern landscape. The specific objectives of our study were to – (i) generate categorical maps of mineral vs. peat soils

across Sweden using multiple definitions of organic layer thickness for peatlands as described above, (ii) produce a continuous organic layer thickness map that could visualize and be useful for any definition of peatlands, (iii) evaluate our predicted peatland estimate of Sweden against inventory data and compare with the existing estimates from traditional maps, and (iv) provide the most accurate national estimates of peatland coverage and constrain the uncertainty in the estimates. This study will provide a guide to mapping mineral and peat soils in any northern

boreal region that will be essential for effective ecosystem management and for supporting sustainable development goals related to restoration of degraded land and climate action.



## 2. Method

### 2.1 Study area

Our study area, the whole of the country of Sweden (latitude 55-70° N, longitude 11-25° E) falls in the boreal and
temperate forest region of Northern Europe (Figure 1A). According to satellite data, the land-cover in Sweden is
dominated by forest, covering 69% of the country, followed by agricultural land (9%), open peatland (9%),
grassland (8%), rock outcrops (5%), and urban land (3%) (Schöllin and Daher, 2019). The climate according to
Köppen, is classified as warm summer continental or hemiboreal climates (Dfb) and subarctic or boreal climates
(Dfc) (Beck et al., 2018). There is a notable elevation and precipitation gradient from north to south, and from east
to west of the country, with annual precipitation ranging from 400 to 2100 mm (1961-1990). The soil type in
Sweden is dominated by Podzols, but more complex distribution of Histosols, Gleysols, Arenosols and Regosols
are also common (Olsson, 1999). The topogenous fens are most common wetland types in Sweden, followed by
string mixed mires and string flark fens (Gunnarsson and Löfroth, 2009).

### 2.2 Swedish Forest Soil Inventory (SFSI)

The Swedish Forest Soil Inventory (SFSI) was used for organic layer thickness data (Olsson, 1999; Stendahl et
al., 2017). The spatial density of the inventory plots varies throughout Sweden due to landscape heterogeneity,
emanating from both natural and human-induced conditions (Figure 1B). The SFSI is conducted on plots with a
radius of 10 m. In case of heterogeneity inside the plots, they are divided into partial plot areas and data are
recorded on the sub plots. In these plots, the organic layer thickness was directly recorded from soil pits (a soil
sampling circle with 1 m radius which is located within the plot area). We included a total of 5 479 data points for
organic layer thickness (Figure 1B).



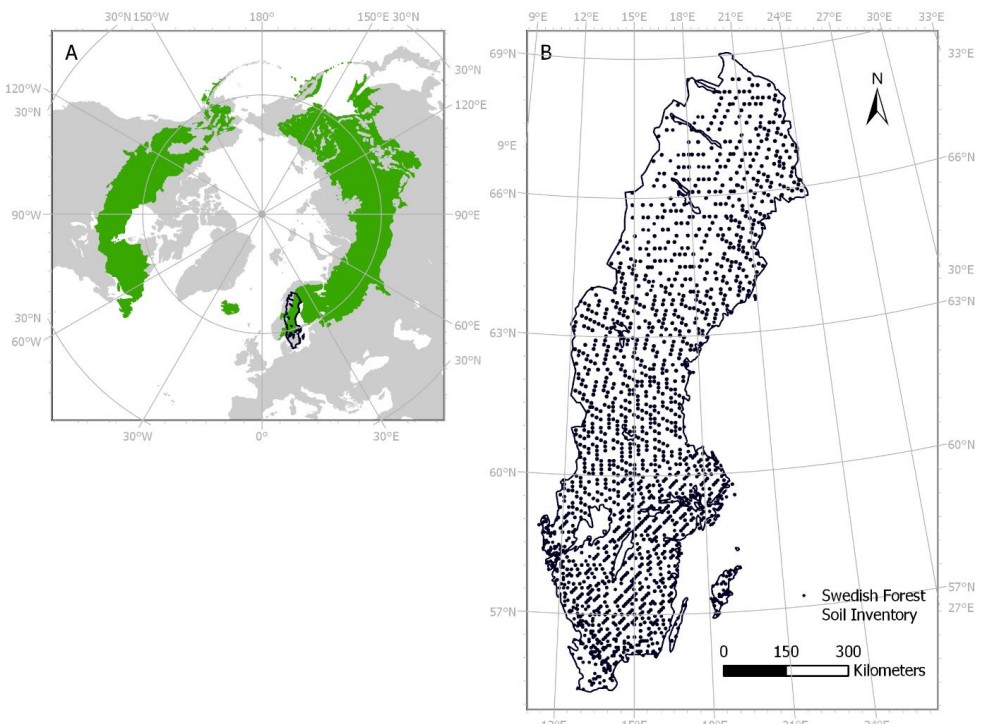

**Figure 1: A) Sweden's position in the northern boreal zone/taiga; map data from (Dinerstein et al., 2017). B) Black dots indicate the sites for the Swedish Forest Soil Inventory (Olsson, 1999; Stendahl et al., 2017) where the thickness of the**
**organic layer has been measured (n = 5 479).**

### 2.3 Generating categorical (peat vs. mineral soils) and continuous organic layer thickness maps

This study utilized the SLU soil moisture map that exhibits soil moisture variation across Sweden on an arbitrary scale from 1 to 100 i.e. from dry to wet (Ågren et al., 2021). It was developed using a combination of digital terrain indices (derived from 2 m resolution digital elevation model based on airborne laser scanning data (ALS) and

ancillary data on quaternary deposits, soil depth, annual and seasonal runoff etc. In total, data from 24 different maps were input into a machine learning model (e.g. Extreme Gradient Boosting model, (Chen et al., 2020)) to predict the soil moisture across Sweden and is now publicly available (www.slu.se/mfk) (Ågren et al., 2021). The soil moisture map has a Cohen's Kappa and Matthews Correlation Coefficient (MCC) values of 0.69 and 0.68, respectively. The map displays the probability of a soil being wet (0-1), which was rescaled to 1-100 (Figure 2A)

so the variability could be displayed without the use of decimals which reduced file size. There is a strong correlation with the probability of a soil being wet and the soil moisture (i.e. Fig. 6 in (Ågren et al., 2021)) and the use of a scale from 1-100 allows the modelling of smooth transitions from dry to wet instead of fixed categories. The current SLU soil moisture map version includes 98.7% of the Swedish landmass. The remaining 1.3% was not laser scanned at the time of map production. Thus, the prediction of peat soil maps in this study also excludes

those 1.3% areas of Sweden, which added minor uncertainties in our national estimates of peat soils.



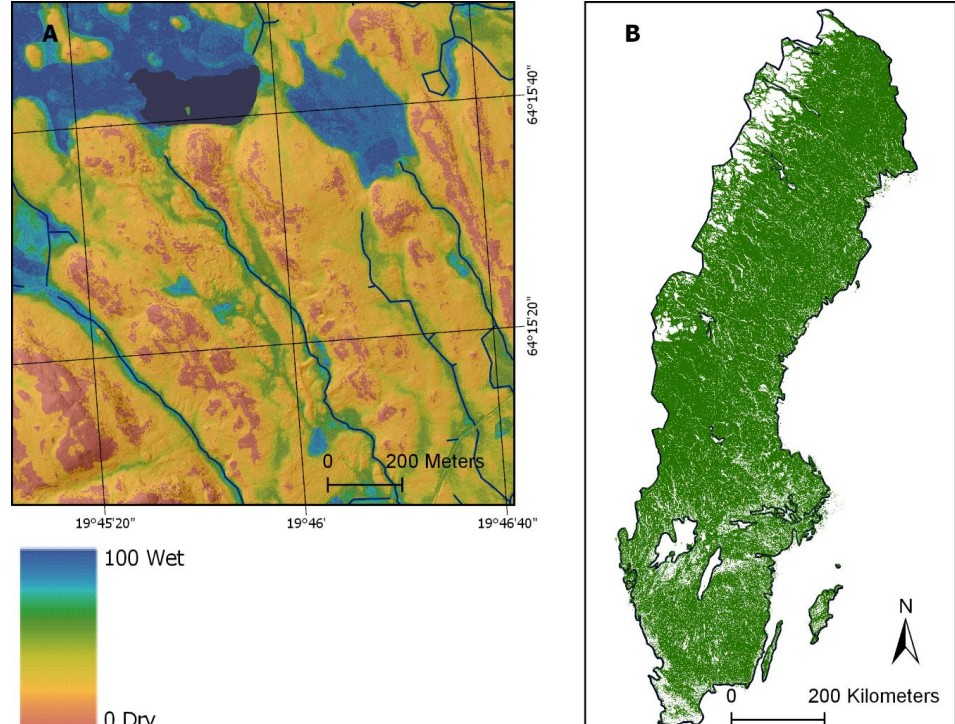

**Figure 2: A) Example of the SLU soil moisture map showing the probability of a soil being wet (0-100%) (Ågren et al., 2021). B) Green areas indicate the forest landscape in Sweden that is sampled by the Swedish National Forest Inventory**
**(Fridman et al., 2014) and the Swedish Forest Soil Inventory (Olsson, 1999; Stendahl et al., 2017).**

We tested the relationship between the SLU soil moisture map and the thickness of the organic layers. The organic layer thickness was registered in the Swedish Forest Soil Inventory up to a maximum thickness of 99 cm. We first divided the Swedish Forest Soil Inventory data (from Section 2.2) into calibration and validation datasets using a
randomized 50% split on IBM SPSS Statistics program. The calibration dataset was then used to establish a relationship between the SLU soil moisture map and the thickness of the organic layers using the curve estimation procedure in IBM SPSS Statistics version 27, which fits a total of 11 linear and non-linear models. The model with the highest $R^2$ was selected to describe the relationship (Figure 3).

The categorical maps were generated based on the relationship with the highest $R^2$. We then tested three different thresholds, i.e. 30, 40, and 50 cm, of organic layer thickness for mapping the peatland extent while the remaining soil was delineated as mineral soil. Hence, three different peatland maps were derived, which we referred as "peat ≥ 30 cm", "peat ≥ 40 cm", and "peat ≥50 cm". In addition, a continuous organic layer thickness map was generated by applying the continuous soil moisture map into the developed model described above. This continuous map
does not contain discrete classes of mineral and peat soils, rather, it presents the distribution of organic layer thickness across the landscape. The accuracy of the maps was then tested using the validation dataset described in Section 2.5.





As the data underlying the maps comes from NFI and SFSI, we lack evaluation data from other land use types.
We therefore first defined the Swedish forest landscape to indicate where the predicted maps could be trusted. The Swedish Forest Soil Inventory samples both productive forest land (defined as areas with a potential wood yield capacity of $> 1$ m$^3$ ha$^{-1}$ yr$^{-1}$ ) and low-productivity forest land (with lower yield capacity), such as pastures, thin soils, non-forest peatlands, rock outcrops, and areas close to the tree line (e.g. birch forests in the alpine region). We generated a map of the Swedish forest landscape (Figure 2B) by reclassifying the National Land Cover
Database (NMD), a land cover map over the entire country in 10 m resolution (Olsson and Ledwith, 2020) where we excluded areas outside the NFI's sampling (crop fields, urban areas, roads, rail roads and power lines) and the alpine region above the birch forest (based on an elevation threshold which is a function of latitude).

### 2.4 Quaternary deposits maps and topographical maps

We used the quaternary deposit and topographic maps of Sweden for comparison with our predicted estimates of
peat soil distribution. In the quaternary deposits map from the Geological Survey of Sweden (SGU), peatland is delineated based on these criteria – (i) organic layer thickness $\geq$ 50 cm, (ii) minimum detected area of 2500 m$^2$, and (iii) the estimated position accuracy ranging 25-200 m (Karlsson et al., 2021). Some linear peatlands narrower than 50 m, but important for the understanding of the geology, were also included and expanded to 50 m width (Pers. comm., C. Karlsson, SGU). However, there are various scales with different coverages for the quaternary
deposit maps in Sweden, such as 1:25 000 (covers 1.7%), 1:50 000 (covers 2.7%) 1:100 000 (covers 47%), 1:200 000 (covers 1.4%), 1:250 000 (covers 21.2%), 1:750 000 (covers 33.6%) and 1:1 000 000 (covers 100%). These maps were merged together to produce a single quaternary deposits map for the whole country where the map with the highest scale was always chosen in areas with overlapping maps (Lidberg et al., 2020). This quaternary deposit map contains five categories of deposit, including till soils, thin soils and rock outcrops, peat, coarse sediments
(sand-gravel-boulders), and fine sediments (clay-silt). The coverage of each category was calculated by summarizing the areas of all polygons within the respective category. Finally, the total coverage of peat category was used for comparison with our predicted estimate of peatland.

Another commonly used mask for delineating peatlands in Sweden is wetlands from the topographic map, i.e. the
Swedish property map (1:12 500) (Lantmäteriet, 2020). However, the wetland class in the property map is not based on the thickness of organic layer; instead, it is defined as peat-forming mires or watery mires and grouped into two categories – (i) wetlands that can be crossed on foot and include mires with shrubs, sedge, and trees of variable densities and (ii) impassable wetlands that are inaccessible on foot and include watery mires, which are mostly fens, soft bed without vegetation, and overgrown lakes with reed. Here we calculated the total coverage of
the wetlands and impassable wetlands by summarizing the areas of the polygons for all of Sweden. These two categories of wetlands were merged together to find a peatland coverage for the whole country and compared with our predicted estimate.

### 2.5 Statistical evaluation of the accuracy of the different peat maps

We evaluated the accuracy of the categorical peatland maps using confusion matrix and the validation dataset (see
Section 2.4). Three predicted peatland maps (i.e. "peat $\geq$ 30 cm", "peat $\geq$ 40 cm", and "peat $\geq$ 50 cm") along with the peatland coverage derived from quaternary deposit and topographic maps were evaluated following the same





approach. More specifically, the ground truth for peat were the SFSI evaluation plots where observed organic layer thickness was larger than the respective thresholds; and the mineral soil ground truth were the SFSI evaluation plots with observed organic layer thickness lower than the respective thresholds. The following accuracy metrics

were calculated based on the confusion matrix:

$$Accuracy = \frac{TP+TN}{TP+FP+FN+TN} \tag{1}$$

$$Precision = \frac{TP}{TP+FP} \tag{2}$$


$$Recall = \frac{TP}{TP+FN} \tag{3}$$

$$Specificity = \frac{TN}{TN+FP} \tag{4}$$

$$MCC = \frac{(TP \times TN) - (FP \times FN)}{\sqrt{(TP+FP) \times (TP+FN) \times (TN+FP) \times (TN+FN)}} \tag{5}$$

$$Kappa = \frac{P_o - P_e}{1 - P_e} \tag{6}$$

Where, True Positives ($TP$) – is the number of observations where the field data and map agrees that soils are peat;
True Negatives ($TN$) – is the number of observations where the field data and map agrees that soils are mineral soils; False Positives ($FP$) is the number of observations where the map predicts peat while soils are mineral soils; False Negatives ($FN$) is the number of observations where the map predicts mineral soils while soils are peat; $P_0$ = Relative observed agreement; $P_e$ = Hypothetical probability of chance agreement. Additionally, the continuous map of the organic layer thickness was evaluated by calculating the goodness of fit ($R^2$) and root mean squared
error (RMSE) from the predicted and observed organic layer thickness.

### 2.6 Peatland estimates from the NFI and SFSI

To compare the predicted peat soil estimates from the maps with other estimates of peatland coverage in Sweden, we also calculated peatland coverage by upscaling from the national inventories, i.e. NFI and SFSI (SLU, 2021) to derive a complete coverage for the Swedish forest landscape. From these inventories, peat coverage can be
derived in 6 different ways: 1) peat coverage was registered in the 2016-2020 NFI survey plots (7 and 10 m radius) in the following classes –  peat coverage 0% (n=33 161), 0-50% (n=1 553), 50-100% (n=1 439), and 100% (n=6 080). For the upscaling, the peat coverage ranges of 0-50% or 50-100% were assumed to cover 25% and 75%, respectively, of the plot. It should be noted that, isolated peatland patches smaller than 25 m$^2$ on plots were disregarded. Details on the NFI data upscaling approach can be found in (Hånell, 2009). 2) NFI also conducts
assessment of cover of different species on 5.64 m plots. On natural peatlands the bryophytes are dominated by the genus *Sphagnum* or Brown mosses (*Amblystegiaceae* family). *Polytrichum commune* commonly also grows in bogs and in riparian zones. Their coverage is measured in the NFI, here we pool the coverage of *Sphagnum*, brown mosses and *Polythrichum commune* into a class that we call "peat indicative mosses". In addition to NFI, the 2003-





2012 SFSI database registered peat soils as 3) quaternary deposits with organic layer thickness ≥ 50 cm and as 4)

255     soil type histosol with organic layer ≥ 40 cm (WRB, 2015). Moreover, SFSI classified the humus form according

to the depths of the OF (Soil Taxonomy Oe), OH (Soil Taxonomy Oa) and H horizons, and amount of aggregates

in case of an A horizon. Based on humus form, peat soils were registered as 5) peat if organic layer ≥ 30 cm and

6) peaty mor if organic layer < 30 cm. This upscaling from the SFSI-database were performed following the

approach described in (Nilsson et al., 2018). The survey data from NFI and SFSI is based on sampling of the

260     Swedish forest landscape, not the total land area. In short this includes productive forest land, pastures, mires, rock

outcrops, alpine region below treeline, but excluding arable land, alpine region above treeline, railroads, power

lines, roads and urban areas. However, the exact definitions of forest land differ slightly among sources which

introduces an uncertainty in the national estimates. For example, the forest landscape mask (Figure 2B) cover

343 000 km$^2$, while NFI and SFSI suggests that 338 000 km$^2$ and 306 000 km$^2$, respectively, are forest land. This

can explain some smaller discrepancies between different sources in Table 2 & 3.

**3. Results**

**3.1 Relationship between soil moisture and thickness of organic layer**

The relationship between the soil moisture variation from the SLU map (Ågren et al., 2021) and organic layer

thickness derived from the Swedish Forest Soil Inventory was well described with a cubic relationship (Eq. 7, R$^2$

= 0.66, p < 0.001; Figure 3).

$Y = 6.4145 + (0.6673 * X) + (-0.0214 * X^2) + (0.0002 * X^3)$            *(7)*

Where *Y* is the thickness of the organic layer (cm) and *X* is the soil moisture level from the SLU soil moisture map.

The s-shape of the curve is due to a rapid increase in organic layer thickness at the high end of soil moisture and a

sharp decrease in organic layer thickness at the lower end of the moisture spectrum. The driest sites are generally

found on crests and ridges characterized by rock outcrops with very thin organic layers. By solving Eq. 7 for *X*

when *Y* was 30, 40 and 50 cm we could determine the soil moisture limits for classifying peat soil. At organic

layer thickness of 30, 40, and 50 cm, the soil moisture limits were 76%, 83% and 87%, respectively.





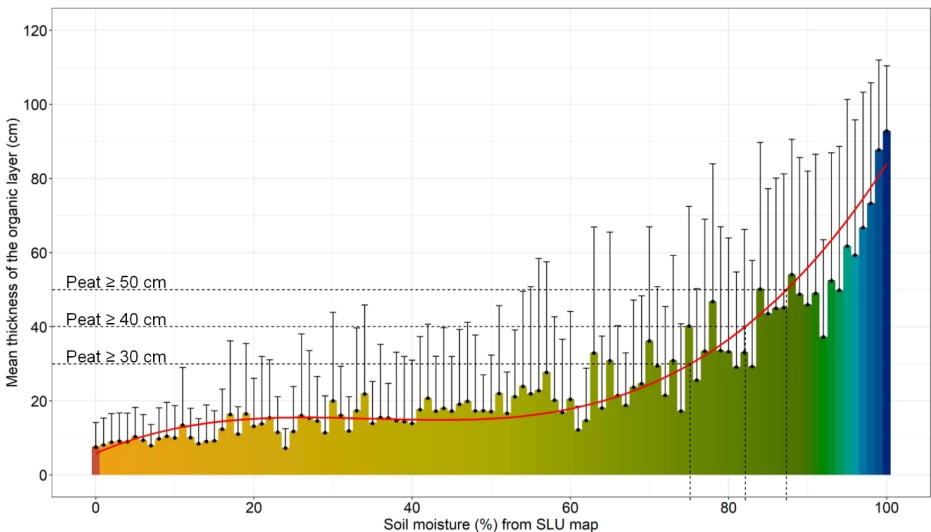


**Figure 3: The red line depicts the cubic relationship ($R^2 = 0.66$) between the thickness of the organic layer in SFSI plots and the probability for the soil being wet according to the SLU soil moisture map (Ågren et al., 2021). The colored bars and black dots indicate the mean organic layer thickness (in the calibration dataset) for soil moisture variation from 0 (dark orange) to 100 (dark blue) as extracted from the SLU soil moisture map (shown using the same color code). The**
**error bars represent the standard error of mean of the organic layer thickness at each soil moisture level. Note that the soil moisture in percentages denotes the probability of a soil being wet expressed as percent, not the volumetric soil water content.**

### 3.2 Statistical evaluation of different peatland maps

Our predicted peat maps generally performed better than the existing topographic and quaternary deposit map products (Table 1). Particularly, the predicted peat ≥50 cm map was of highest quality in terms of accuracy, recall, kappa, MCC values. The prediction of peat ≥40 cm and peat ≥30 cm had equal kappa (i.e. 0.69) and MCC (i.e. 0.69) values; however, they were about 5% less accurate than the kappa and MCC of peat ≥50 cm. Although all three predicted peat maps exhibited better accuracies for most metrics, the topographic map had higher precision
and specificity values.




**Table 1: Evaluation metrics of different maps of peat soils. TP = True Positives, TN = True Negative, FP = False Positives, FN = False Negatives. The other metrics are explained in Eq. 1-6. Kappa refers to Cohen's kappa and MCC to Matthews's correlation coefficient. The peat maps that we predicted in this study are highlighted in italics. The topographic and quaternary deposits maps are existing national products.**

|  | TP | TN | FP | FN | Accuracy (%) | Precision (%) | Recall (%) | Specificity (%) | Kappa | MCC |
|---|---|---|---|---|---|---|---|---|---|---|
| *Peat ≥50 cm map* | *427* | *2217* | *135* | *104* | *91.7* | *75.9* | *80.4* | *94.3* | *0.73* | *0.73* |
| *Peat ≥40 cm map* | *417* | *2133* | *165* | *114* | *90.1* | *71.7* | *78.5* | *92.8* | *0.69* | *0.69* |
| *Peat ≥30 cm map* | *537* | *2013* | *199* | *134* | *88.5* | *72.9* | *80.0* | *91.1* | *0.69* | *0.69* |
| Topographic map | 264 | 2318 | 39 | 266 | 89.4 | 87.1 | 49.8 | 98.4 | 0.58 | 0.61 |
| Quaternary deposits map | 363 | 2227 | 130 | 167 | 89.7 | 73.6 | 68.5 | 94.5 | 0.65 | 0.65 |

The prediction of continuous organic layer thickness captures the general patterns, with a positive relationship between measured and observed thickness of organic layer ($R^2$ of 0.67 and p<0.001). However, the confidence interval (CI) in Figure 4 and a root mean square error (RMSE) of 19 cm, indicate a rather large uncertainty in the estimated thickness of the organic soils. Moreover, the cubic relationship shown in Figure 3 (i.e. Eq. 7) could not fully capture the rapid increase in organic layer thickness that occurred with high soil moisture and the sharp decrease in organic layer thickness with dry soils. Hence, the predictions in Figure 4 only ranged 6-95 cm compared with the measured data which ranged 0-99 cm. As a result, the model systematically overestimates thin organic layers and underestimates thick organic layers.

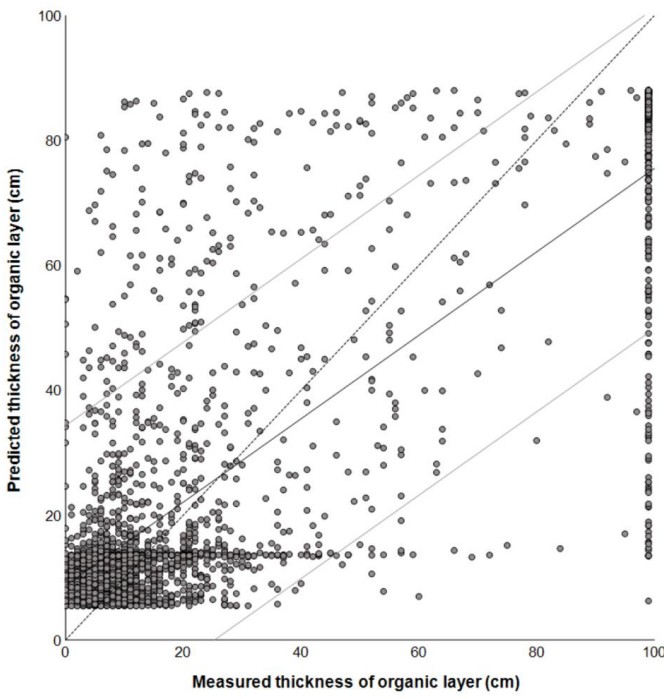



**Figure 4: Predicted vs measured continuous thickness of the organic layer in the evaluation dataset (n = 2 883). Dashed line indicate the 1:1 line and black line a linear regression (R² = 0.67, p<0.001), grey lines indicate 95% CI. Field measurements of peat thickness that were 99 cm or above were reported as 99 cm, hence the many over lapping data points.**

### 3.4 Viewing different peatland maps

As described above, the predicted area of peat ≥ 50 cm covers larger areas than the existing national maps (Table 2), which was also evident in Figure 5 (panels A and D). The peat ≥ 50 cm map captured more riparian peat soils than the topographic and quaternary deposit maps (Figure 5A). It also better delineated the mire areas that are obscured by tree canopy and typically not captured using traditional mapping techniques based on aerial photos (illustrated in Figure 5D, c.f. hillshade and aerial photo in Figure A1 in Appendix A). Although there were differences in peatland coverage between the predicted maps at different thresholds (i.e. ≥30, ≥40 and ≥50 cm peat maps), they provided more or less comparable distribution of peat across the landscape (Figure 5 B, E). A RMSE of 19 cm for the prediction of continuous organic layer thickness (Figure 5 C, F), indicate that the depth estimates are uncertain, and should not be taken literally. However, we argue that this map can be used to display the horizontal distribution of peat and mineral soils and indicate a smoother and more realistic impression of the high variability in areal distribution of the organic soils. The map with continuous organic layer depths exhibits pixel-by-pixel variation in organic layer thickness ranging from 6 to 95 cm across the country and unlike the categorical maps, does not demonstrate discrete soil classes which may cause misrepresentation of natural conditions and distribution of organic soils due to oversimplification.



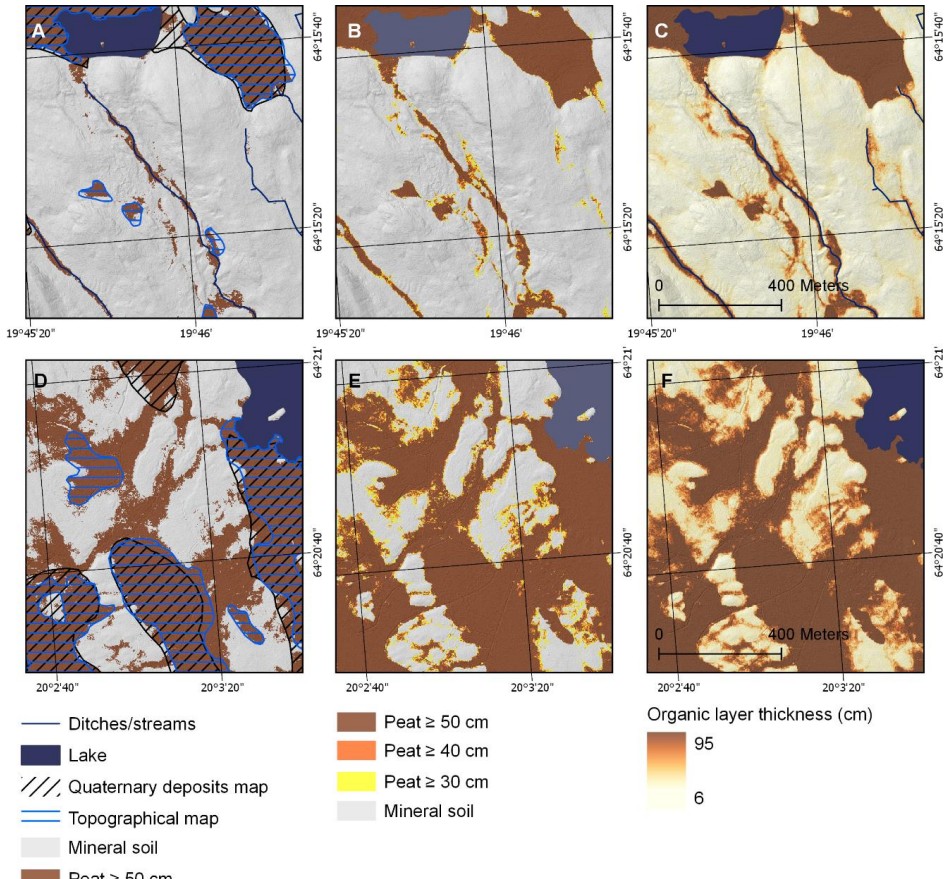

**Figure 5: The left panels (A, D) show an example of two different locations with peatlands with the following three maps overlaid on each other for comparison: quaternary deposits map (1: 25,000) (black hatched area), topographical map (1: 12 500) (blue hatched area) and the predicted peat ≥ 50 cm map (2 m resolution) (brown area). The centre panels (B, E) demonstrates the difference in the predicted peat maps using 30, 40 and 50 cm organic layer thresholds by superimposing the predictions on top of each other with peat ≥ 30 cm in the bottom (a wider distribution) and peat ≥ 50 cm at the top (a narrower distribution). The right panels (C, F) exemplifies the continuous map of organic layer thickness derived using the Eq. 7 in two different areas. Note that the thickness of the organic layer may be underestimated in some areas as the full depth of the organic deposits are not registered in the Swedish Forest Soil Inventory database (ranges 0-99 cm).**

## 3.5 Peat coverage in Sweden

We observed notable differences in peat coverage between our predicted maps and estimates from the existing map products (Table 2). There was large variation in our predicted estimates of peatland coverage at different thresholds of organic layer thickness ranging from 70 000 – 94 000 km² and suggests that 18 – 24% of Swedish landmass is covered by peatland, depending on the definition used, which is considerably larger than the estimates from the existing Swedish map products. Namely, the peatland coverage from topographic map was only 13%, while it was just 14% based on the quaternary deposit map. This is in comparison to the coverage of other





quaternary deposits in Sweden from the calculations of the quaternary deposits map: till soils 53%, thin soils and rock outcrops 18%, coarse sediments (sand-gravel-boulders) 8%, fine sediments (clay-silt) 6%, and other (ice, fillings, etc.) 1%. The forest landscape, according to the map in Figure 2B cover ca 85% of the land area in Sweden.

While excluding peatlands outside forest land decreased the overall coverage of peat to 68 000 - 88 000 km$^2$, we see that in relative terms, peatlands were more common in the forest landscape (21-26%) (Table 3) compared to the national averages (Table 2). The peat coverage estimates of the forest landscape according to upscaling from NFI and SFSI (section 2.6) ranged 55 000-91 000 km$^2$, depending on the definition used. The average errors from these surveys range 2-4%.


**Table 2: Coverage of peatlands in Sweden according to different maps, in km$^2$ or in % of the land area in Sweden (excluding lakes and large ≥ 6m wide rivers). The peat maps that we predicted in this study are highlighted in italics. The topographic and quaternary deposits maps are existing national products.**

| Map | Peat coverage of total land area | |
|---|---|---|
| | (km$^2$) | (%) |
| *Peat ≥50 cm map* | *70 000* | *18* |
| *Peat ≥40 cm map* | *79 000* | *20* |
| *Peat ≥30 cm map* | *94 000* | *24* |
| Topographic map | 56 000 | 13 |
| Quaternary deposits map | 58 000 | 14 |


**Table 3: Peatland coverage in km$^2$ or in percent of forest land according to different sources. The peat maps that we predicted in this study are marked in italics. The numbering refer to the different upscaling estimates from inventory data as described in section 2.6. Forest land includes productive forest land, pastures, mires, rock outcrops, alpine region below treeline, but excluding arable land, alpine region above treeline, railroads, power lines, roads and urban**
**areas. However, the exact definitions differ slightly among sources which can explain some inconsistences in the % cover among the three data sources in the table (maps, NFI and SFSI data), see section 2.6.**

| Source | Peat coverage of forest land area | |
|---|---|---|
| | (km$^2$) | (%) |
| *Peat ≥50 cm map* | *68 000* | *21* |
| *Peat ≥40 cm map* | *76 000* | *23* |
| *Peat ≥30 cm map* | *88 000* | *26* |
| 1. Peat coverage ≥30 cm according to upscaling from NFI | 65 000 | 19 |
| 2. Peat indicative mosses according to upscaling from NFI | 65 000 | 19 |
| 3. Peat coverage ≥ 50 cm according to upscaling from SFSI | 55 000 | 18 |
| 4. Peat coverage ≥ 40 cm according to upscaling from SFSI | 60 000 | 20 |
| 5. Peat coverage ≥ 30 cm according to upscaling from SFSI | 63 000 | 20 |
| 6. Peat coverage ≤ 30 cm according to upscaling from SFSI | 91 000 | 30 |
| **Average (Standard Deviation)** | **71 000 (±12 000)** | **22 (±4)** |



**4 Discussion**

Using Sweden as a test case, this study provides a guide to improved mapping of peat and mineral soils using ALS
        data across large areas – that can be applied to other boreal forest regions. We have successfully shown in a largely
        boreal landscape that we can use soil moisture to predict spatial distribution of peat soils more accurately than
        previous techniques used for the existing national maps. Specifically, these new maps include smaller areas with
        peat or tree covered peat soils previously overlooked in earlier maps. This new map of peat soils was developed

to support the need for land use management optimization, by incorporating landscape sensitivity and hydrological
        connectivity into a framework that promotes a rational and sustainable management of organic and wet soil areas.
        Improved decision-support tools hold the key for land-use management policies, and as we enter the UN Decade
        on Ecosystem Restoration, mechanistic insights into restoration targets become increasingly important. For
        example, the peat maps can be used to plan land-use management such as planning road constructions or off road

driving, designing riparian protection zones to optimize the protection of water quality and biodiversity, or guiding
        the restoration of drained wetlands. According to Minasny et al. (2019), global estimates of soil C stocks have
        improved over the last decade (Arrouays et al., 2014). But, as digital maps of peatlands are typically of low quality
        globally, C stock estimates for peatlands vary considerably, between 113 and 612 Pg (Jackson et al., 2017).
        Improved mapping of peatlands can therefore also answer more fundamental research questions such as improving

future estimates of soil carbon stocks.

**4.1 Categorical maps - delineation of peat soils**

        Our categorical peat maps based on predictions from soil wetness were of substantially higher quality compared
        to the spatial peat distribution from the existing national topographic and quaternary deposits maps (Table 1). In

fact, all evaluation metrics except accuracy and specificity measures (Table 1) were higher for our predicted
        categorical peat maps. The slightly lower accuracy and specificity of the predicted maps might result from the
        unbalanced evaluation dataset. In our binary categorical map, peatland coverage (13-24% of total land/forest area)
        was significantly less than the mineral soils coverage (76-87% of total land/forest area); hence, the evaluation
        dataset had more data points for mineral soils resulting in slightly higher accuracy and specificity for the traditional

maps. However, for such unbalanced evaluation data, kappa and MCC measures are better metrics for overall
        prediction quality than accuracy, which can give overoptimistic results driven by the larger class (i.e. mineral soil
        in our case) while we are in fact more interested in the smaller class (i.e. peat soil) (Delgado and Tibau, 2019;
        Chicco and Jurman, 2020).

Our predicted peat maps captured smaller peat areas as small as 4 m$^2$ due to the high quality input data of 2 m
        spatial resolution (Figure 5A), while the existing traditional maps only include peatlands larger than 2 500 m$^2$. Our
        peatland maps can therefore identify peat soils along stream networks as riparian peat or in smaller pockets in the
        bottom of small valleys (Figure 5A) where groundwater flow paths converge. These more local scale peat soils
        are common in the boreal region, and are sometimes called "cryptic wetlands" (Creed et al., 2003), discrete riparian

input points (DRIPS) (Ploum et al., 2018) or groundwater discharge areas (Kuglerova et al., 2014a) and are more
        connected to mineral soils. Such areas often have higher nutrient status and pH and more nutrient demanding plant
        species (Kuglerova et al., 2016; Kuglerova et al., 2014b; Rydin et al., 1999) than larger mire complexes. In





addition, we noted that peat soils seemed to be under-estimated in the forested areas in the traditional maps. Black and white aerial photos (or color and IR-orthophotos more recently) were used for the delineation of peatlands in

traditional mapping in combination with field observations - mainly along the roads. As a result, the cartographers interpreted many areas under dense forest canopy as mineral soils, a common misinterpretation when mapping soils from aerial photos. A typical example of such cartographic challenges are provided in Figure 4D and Appendix A. The flat low laying areas drained by ditches (Appendix A, Figure 1C) is a forested peatland area (Appendix A, Figure 1D), that was misclassified on traditional maps (5D). Therefore, such traditional mapping

techniques have likely resulted in underestimation of productive, now forested peatlands. Our peat maps, based on predictions from soil wetness, were mainly based on digital terrain indices and high-resolution laser scanning data (Ågren et al., 2021) that are not restricted by dense forest canopies and thus, could provide much more accurate estimates of peat soils. Our predicted maps therefore capture larger areas of peat soils previously unmapped and had a recall rate in the order of 80% that can be compared to ca 50% and 70% on topographic and quaternary

deposits maps, respectively.

While the peat ≥ 50 cm map had the highest overall quality, all of the predicted maps (i.e. peat ≥30 cm, ≥40 cm and ≥50 cm) were qualitatively more or less comparable, at least in comparison to traditional maps (Table 1, Figure 5 B, E), i.e. the spatial overall distribution remained, even if the area covered by peat soils increased when moving

from ≥50 cm to ≥30 cm peat depth. The delineation of peat soils using peat that is 30, 40 or 50 cm can be valuable depending on the research or management objectives. Even though the error bars in Figure 3, and an RMSE of 19 cm for the continuous map of the organic layer thickness, indicate that there is some level of uncertainty for the estimates of peat soil depth, and should not be taken literally, the categorical maps still delineated peat soils better than traditional maps (i.e. quaternary deposits maps and topographical maps).


### 4.2 Map of the continuous thickness of the organic layer

Categorization is a fundamental mechanism of human cognitive construction (by dividing the subject of interest into groups and comparing them, we form our knowledge of the world) (Mcgarty, 2015). Such categorical divisions may however cause overgeneralization and inaccurate representation of the true distributions. We argue

that in nature, there is often a more gradual shift from mineral soil to peat soil, rather than a clearly defined border. Applying a cubic relationship (Eq. 7), we could model the thickness of the organic layer from the SLU soil moisture map. The high-resolution (2 m) SLU soil moisture map, displaying the probability of a soil being wet, captures the gradual shifts in soil moisture across the natural landscape (Figure 2). The high quality of the SLU soil moisture map is obtained by combining data from 24 different spatial data sources in a machine learning model (i.e. Extreme

Gradient Boosting (Chen et al., 2020)) to adjust the map to both regional and local conditions based on the observations from ~16 000 National Forest Inventory (NFI) plots across Sweden. Ågren et al (2021) found that the SLU soil moisture map captures 79% of wet soils, suggesting a significant improvement over the existing map products. Hence, the SLU soil moisture map has enabled the possibility of predicting how water follows the flow paths from each ridge into local valleys where the groundwater is concentrated in swales (i.e. cryptic wetlands and

riparian peats), to further downstream into flat areas where water gets stagnant with high groundwater-levels typically landscapes with mixed mire complexes. We have now shown that this continuum of hydrological



connectivity of the landscape has a significant relationship with the organic layer thickness, and thus, can be effective for tracking the distribution of the organic layer thickness in a continuous map (Figure 5C, F). In the study region, most organic soils are overlaying till deposits, which are highly heterogeneous and often anisotropic.

The surface roughness of the underlying till will have a local effect on peat depth, which likely contribute to the relatively high RMSE for the continuous peat depth map. In short, the maps are based on modelling from soil surface data from ALS measurements, while stones, boulders or ridges made of till can be hidden below the flat peat surface, affecting the peat thickness (Nijp et al., 2019). The relatively high RMSE of 19 cm, is also an indication that the delineations of peat soils based on a defined thickness of the organic layer (i.e. ≥30, ≥40 or ≥50

cm) should not be taken literally. We therefore argue that a map based on continuous organic layer thickness (Figure 5C, F) provides a more realistic representation of peat soil distribution in the natural landscape and comprises a better basis for addressing specific research or management questions. It should be noted that the map will not capture the full depth of the peat deposits, however, this was not the purpose of this mapping analysis. Mean peat thickness in Sweden has been estimated to be 1.52 m in north Sweden, 1.94 m in south-central Sweden

and 2.26 m in south Sweden (Franzen et al., 2012). Therefore, the depth of the organic layers should not be taken literally, but the continuous map can be used to indicate the horizontal distribution of peat soils instead of using a fixed threshold. Light yellow areas on the continuous map are indicative of mineral soils, and brown areas are indicative of peat soils, while the areas that show a rapid change in colour are indicative of the transition zone between mineral soils and peatlands (Figure 5 C & F). We argue that this is an intuitive way of illustrating the

uncertainty in the borders between mineral and peat soils.

### 4.3 National estimates of peat coverage for the forest landscape

Our new estimations of peatland coverage for all of Sweden ranged 70 000-94 000 km$^2$ and are better than previous estimates from quaternary deposits map (58 000 km$^2$) and topographical maps (56 000 km$^2$) (Table 2), given that

the maps of ≥50, ≥40 and ≥30 cm peat had a higher quality (Table 1). For the first time, it is possible to produce maps that delineate each individual peat deposit, and that give more reasonable estimates of the national peat cover for Sweden. The new maps produce peat estimates for Sweden close to some of the best estimates, based on a combination of data sources and upscaling; 85 023 km$^2$ (Barthelmes et al., 2015) and 63 700 – 69 200 km$^2$ (Tanneberger et al., 2017). While it is interesting to compare the peat coverage for the national estimates, the

predicted maps can only be trusted for the forest landscape, as our study is based on sampling of the forest landscape. The peat coverage estimates will vary depending on the definition used (Table 3), and all sources have uncertainties. However, by calculating several measures we can constrain the national estimates of peat soil coverage on Swedish forest land. In general, there was a strong agreement in the total peat soil area derived from soil wetness based predictions and from statistical upscaling of the national inventories. The predicted maps from

soil moisture has slightly larger areas of peat coverage (77 000 ±10 000 km$^2$) compared with the estimations from the NFI and SFSI surveys (67 000 ±14 000 km$^2$), but lower than earlier estimates from NFI; 83 000 km$^2$ (Hånell, 2009) and 100 000 km$^2$ (including peaty mor) (Hånell, 1990). One potential explanation why the maps predict larger areas than peat coverage from NFI can be that isolated peat soil patches smaller than 25 m$^2$ are disregarded in the NFI survey, while our maps predict peat areas as small as 4 m$^2$. It can therefore be assumed that many of

the smaller peatland features such as riparian peat or peat in local pits/swales to a certain degree are disregarded



in the NFI data but are included in our new map predictions. Given an RMSE of 19 cm for the continuous map, we argue that one should not analyse different peat depth maps estimates in detail, rather all the different estimates can be used to constrain the uncertainty in the peatland estimates. The peat coverage of Swedish forest land according to all estimates is 71 000 ±12 000 km$^2$ (22 ±4%). In addition, on agricultural land the total area of peat

soil used in agricultural production is estimated to be 2 257 km$^2$ (7% of the total agricultural area) of which 80% is used as arable land and 20% for pasture (Minasny et al., 2019). Another estimate of agricultural peat and gyttja soils in Sweden is 3 015 km$^2$ (Berglund and Berglund, 2010). Furthermore, the alpine region above the birch forest is estimated to have 3 040 km$^2$ of peatlands, but these numbers are uncertain due to few observations (Löfgren, 1998). A clear advantage with our method of mapping peatlands compared with previous maps generated from

NFI data is that our new method for the first time allows for a delineation of all peat soils across Sweden, while maps from NFI data traditionally only show statistical fractions of peat coverage per land area at county scale, i.e. not spatially explicit distribution (Nilsson et al., 2001; Olsson, 1999).

### 4.3 The novelty of the developed maps

A recent review of digital mapping of peatlands show that there has been a successive increase in our ability to

map peatlands globally via digital mapping using remote sensing and satellites such as Landsat, Sentinel and MODIS, or, using satellites that measure earth's surface moisture (Gravity Recovery And Climate Experiment (GRACE) available at about 50 km resolution, and Soil Moisture Active Passive (SMAP) available at 3 km resolution) (Minasny et al., 2019). While the coarse resolution satellite data may be useful for continental or global coarse mapping, it is not adequate for detailed planning of land-use management. Moreover, the review showed

that while it is common to delineate peat extent, studies rarely perform validation or calculations of the uncertainty of the predictions (Minasny et al., 2019). Here, we present a novel way of delineating peatlands across an entire country, at a very fine spatial resolution (2 m), in addition we validate the maps using a separate evaluation dataset and several evaluation metrics (Table 1). Furthermore, we calculate several estimates of the peatland coverage of the Swedish forest landscape which allows us to constrain the estimates (Table 3). Schönauer (2022), recently

showed that combining airborne laser data with other map sources and AI models they could produce accurate soil moisture maps for 6 study areas in Finland, Germany, and Poland. By applying an XGBoost machine learning model for predicting soil moisture they predicted 74% of wet values correctly, a significant improvement compared to depth-to-water-maps that predicted 38% of wet values correctly. As the number of countries that have wide-area public Lidar datasets are increasing in the northern boreal zone (Cohen et al., 2020) and new methods of

mapping soil moisture using machine learning from a combination of data sources (Schönauer et al., 2022; Ågren et al., 2021) are being developed, this study can provide a benchmark for new and improved peatland maps of the northern boreal zone at a nationwide scale. This can bridge an important research gap between global scale mapping using satellites on a coarse scale, and detailed field scale mapping (Minasny et al., 2019).

### Conclusions

An empirical relationship between the thickness of the organic layer and the continuous SLU soil moisture map (R$^2$ = 0.66, p < 0.001) was used to generate 3 categorical maps of peat distribution in Sweden (using peat depths of ≥ 30, 40 or 50 cm respectively as thresholds). The developed peat maps had a higher overall quality (MCC = 0.73) compared to traditional quaternary deposits maps (MCC = 0.65) and topographical maps (MCC = 0.61) and



captured more of the peatlands with a recall of ca 80% compared to 50-70% on the traditional maps. The ability
        to map smaller scale peatlands as fine as 4 m$^2$ and the fact that our predicted peat maps were not restricted by
        dense forest canopies (as our maps were based on high resolution digital terrain indices) together provided better
        estimates of peat soils that nearly doubled the accounting of peat soil areas for Sweden compared to other national
        map products. We also provided a continuous map of the organic layer depth, ranging from 6-95 cm, with an R$^2$
        of 0.67 and RMSE of 19 cm. This continuous map exhibits a smooth transition from mineral to peat soils and

provides an intuitive uncertainty estimate in the horizontal delineation of peat soils. Finally, by calculating several
        measures of peat soils, we can constrain the uncertainties in the national estimates of peat soils in the Swedish
        forest landscape to 71 000 ±12 000 km$^2$ or 22 ±4%.

**Acknowledgements**

This work was funded by Formas (project no. 2019-00173, 2021-00115, 2021-00713, 2018-00723, 2016-00896)
and Knut and Alice Wallenberg foundation (project no. 2018.0259). We would also like to thank the staff at the
        Swedish Forest Soil Inventory for providing the data for this article and Jonas Dahlgren, at NFI who calculated
        the peatland cover from the NFI data (section 2.6).

**Code/Data availability**

The SLU soil moisture map that underlie this study is open data ([www.slu.se/mfk](www.slu.se/mfk)).

**Author contribution**

AÅ: Conceptualization, Methodology, Statistical analysis, Map production, Writing - Original Draft,
Visualization, Funding acquisition. EMH: Writing. JS: Methodology, upscaling estimates, writing. MBN: Writing.
SSP: Methodology, Statistical analysis, Map production, Writing.

**Competing interests**

The authors declare no competing interests.

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



**Appendix A**

Figure A1: Hillshade and aerial photo of the two mapped areas shown in Figure 5. © Lantmäteriet