# Peer review of "Delineating the distribution of mineral and peat soils at the landscape scale in northern boreal regions"

_EGUsphere, 2022_

## Author Comment (AC1)

**Author response to Editor**

**EC1: David Dunkerley, 28 Jun 2022**

This paper presents a study that sought to revise the mapping of peat soils at national scale across Sweden.

The 'Methods' section reports that the primary data used are detailed elevation data from airborne laser scanning (at 2 m resolution), and a computed soil moisture map previously developed by Ågren et al. The soil moisture map in turn relied on unspecified 'digital terrain indices' (ms. line 143-144), together with "ancillary data on quaternary deposits, soil depth, annual and seasonal runoff etc " (ms. line 145) that were used as input for a machine learning model, to predict soil moisture across Sweden. This section of the Methods presentation seemed inadequate to me. 1) What were the topographic indices? 2) How and at what scale were they derived? 3) How were annual and seasonal runoff quantified, and what was the resolution and quality of these data? 4) Runoff data can surely have been at no finer scale than that of catchment level, in most cases. If so, how can it assist in mapping peat at 2 m resolution? The authors need to explain much more thoroughly the data used and the methods used in the machine learning model. 5) In turn, more commentary was needed on the resolution and quality of the soil moisture maps. 6) What, for instance, is the extent of seasonal variability? 7) Is the parameter calculated perhaps an annual mean or median value?

*Author response:*

*Thank you for your feedback. We answered each question below:*

**Question:** What were the topographic indices? How and at what scale were they derived?

***Response:*** *For producing the soil moisture map, we included a total of 45 input variables a.k.a. features, which were mostly derived from ALS data. However, after performing the feature reduction procedure, which is a common procedure of developing machine learning models, a total of 28 features remained in the final model. The variables highlighted in black in the table below were used to derive the SLU soil moisture map.*

*Here, we also report some relevant texts from (Ågren et al., 2021) to answer the question regarding how and at what scale the predictor variables were developed:*

*"The soil moisture and local topography measures were all calculated from the 2 m national DEM, apart from the Topographic Wetness Index (TWI), which has been found to give unrealistic results when calculated at high resolution (Sørensen and Seibert, 2007, Ågren et al., 2014b). Therefore, TWI was calculated at coarser resolutions (10–48 m) deemed sufficient to capture the macro-topographical control of hydrological pathways. By including different window sizes (6 × 6 m to 160 × 160 m) we evaluated both macro- and micro-topographic effects on these pathways (Table 2). However, as we were applying substantially higher resolution than many other studies, it also enabled us to evaluate the modeling utility of more 'small-scale features'. For this purpose we incorporated the following digital terrain indices in addition to those described by (Lidberg et al., 2020)—the downslope index (Hjerdt et al., 2004), standard deviation of mean elevation within a moving window of 7 × 7 DEM cells, standard deviation from slope with a moving window of 3 × 3 cells, circular variance of aspect with a 3 × 3 moving window, and ruggedness index—all calculated from the 2 m DEM. For an explanation of these indices see the WhiteboxTools User Manual (Lindsay,*

_2020_). By including more of these 'small-scale features' we aimed to improve the modelling of soil moisture in local pits and small-scale variability in riparian zones. Ancillary environmental variables used to capture variability in climatic and soil conditions were: quaternary deposits and soil depth from the Swedish Geological Survey; wetlands from the Swedish Mapping, Cadastral and Land Registration Authority; runoff from the Swedish Metrological and Hydrological Institute; and land-use from the national land cover database as well as a 10 m resolution soil moisture index from the Swedish Environmental Protection Agency (SEPA). These data, summarized in (Table 2), were resampled to 2 m grids to match the LIDAR-derived variables."

Table 2. Input variables used to model soil moisture, including digital terrain indices and ancillary environmental variables, calculated as described by [1](Lidberg et al., 2020) and [2](Lindsay, 2020). Abbreviations refer to the designations in Figure 4. Features included in the final model are marked in black and features that were evaluated but excluded from the final model are marked in grey.

| In-data map layers used to classify soil moisture | Utilized scales, thresholds and seasons | Data source | Short name |
|---|---|---|---|
| Soil moisture measures | | | |
| Depth to water 0.5 ha [1] | Stream initiation threshold 0.5 ha | Calculated from the 2 m DEM | DTW 0.5 ha |
| Depth to water 1 ha [1] | Stream initiation threshold 1 ha | Calculated from the 2 m DEM | DTW 1 ha |
| Depth to water 2 ha [1] | Stream initiation threshold 2 ha | Calculated from the 2 m DEM | DTW 2 ha |
| Depth to water 5 ha [1] | Stream initiation threshold 5 ha | Calculated from the 2 m DEM | DTW 5 ha |
| Depth to water 10 ha [1] | Stream initiation threshold 10 ha | Calculated from the 2 m DEM | DTW 10 ha |
| Depth to water 15 ha [1] | Stream initiation threshold 15 ha | Calculated from the 2 m DEM | DTW 15 ha |
| Depth to water 30 ha [1] | Stream initiation threshold 30 ha | Calculated from the 2 m DEM | DTW 30 ha |
| Down slope index [2] | 2 m drop threshold | Calculated from the 2 m DEM | DI 2 m |
| Topographic wetness index [1] | 10 m × 10 m | Calculated from a 10 m DEM resampled to 2 m | TWI 10 m |
| Topographic wetness index [1] | 24 m × 24 m | Calculated from a 24 m DEM resampled to 2 m | TWI 24 m |
| Topographic wetness index [1] | 48 m × 48 m | Calculated from a 48 m DEM resampled to 2 m | TWI 48 m |
| Elevation above stream 0.5 ha [1] | Stream initiation threshold 0.5 ha | Calculated from the 2 m DEM | EAS 0.5 ha |
| Elevation above stream 1 ha [1] | Stream initiation threshold 1 ha | Calculated from the 2 m DEM | EAS 1 ha |
| Elevation above stream 2 ha [1] | Stream initiation threshold 2 ha | Calculated from the 2 m DEM | EAS 2 ha |
| Elevation above stream 5 ha [1] | Stream initiation threshold 5 ha | Calculated from the 2 m DEM | EAS 5 ha |

| | | | |
|---|---|---|---|
| Elevation above stream 10 ha [1] | Stream initiation threshold 10 ha | Calculated from the 2 m DEM | EAS 10 ha |
| Elevation above stream 15 ha [1] | Stream initiation threshold 15 ha | Calculated from the 2 m DEM | EAS 15 ha |
| Elevation above stream 30 ha [1] | Stream initiation threshold 30 ha | Calculated from the 2 m DEM | EAS 30 ha |
| Local topography measures | | | |
| Elevation [1] | Elevation of each field plot | Calculated from the 2 m DEM | Elevation |
| Standard deviation from mean elevation [2] | Standard deviation from mean elevation with a 7 x 7 cell moving window | Calculated from the 2 m DEM | DFME |
| Standard deviation from elevation [1] | Standard deviation from the digital elevation model with a 5 x 5 cell moving window | Calculated from the 2 m DEM | STDV 5 Cells |
| Standard deviation from elevation [1] | Standard deviation from the digital elevation model with a 10 x 10 cell moving window | Calculated from the 2 m DEM | STDV 10 Cells |
| Standard deviation from elevation [1] | Standard deviation from the digital elevation model with a 20 x 20 cell moving window | Calculated from the 2 m DEM | STDV 20 Cells |
| Standard deviation from elevation [1] | Standard deviation from the digital elevation model with a 40 x 40 cell moving window | Calculated from the 2 m DEM | STDV 40 Cells |
| Standard deviation from elevation [1] | Standard deviation from the digital elevation model with a 80 x 80 cell moving window | Calculated from the 2 m DEM | STDV 80 Cells |
| Standard deviation from slope [2] | Standard deviation from slope with a 3 x 3 cell moving window | Calculated from the 2 m DEM | SDFS |

| Circular variance of aspect [2] | Circular variance of aspect with a 3 x 3 cell moving window | Calculated from the 2 m DEM | CVA |
|---|---|---|---|
| Ruggedness [2] | Ruggedness index | Calculated from the 2 m DEM | Ruggedness |
| Slope [2] | Slope | Calculated from the 2 m DEM | Slope |
| Ancillary environmental variables | | | |
| Quaternary deposits - Peat Soil [1] | Extracted from the best map available at each field plot (with scales differing among regions) | Digital map from the Geological Survey of Sweden | Peat Soil |
| Quaternary deposits – Glacial till [1] | -\|\|- | -\|\|- | Till soil |
| Quaternary deposits - Fine sediment [1] | -\|\|- | -\|\|- | Fine sediment |
| Quaternary deposits – Coarse Sediment [1] | -\|\|- | -\|\|- | Coarse sediment |
| Quaternary deposits - Thin soil [1] | -\|\|- | -\|\|- | Thin soil |
| Soil depth | Modelled soil depth | -\|\|- | Soil depth |
| Wetlands from the Swedish property map [1] | 1:12 500 scale | From the Swedish Mapping, Cadastral and Land Registration Authority | Wetlands |
| Winter Runoff [1] | 30-year average winter runoff | Digital map from Swedish Metrological and Hydrological Institute, Calculated with S-HYPE | Winter Runoff |
| Summer Runoff [1] | 30-year average summer runoff | -\|\|- | Summer Runoff |
| Autumn Runoff [1] | 30-year average autumn runoff | -\|\|- | Autumn Runoff |
| Annual Runoff [1] | 30-year average annual runoff | -\|\|- | Annual Runoff |
| Spring Runoff [1] | 30-year average spring runoff | -\|\|- | Spring Runoff |
| X Coordinates of field plots [1] | X coordinates in SWEREFF 99 TM | NFI | X Coordinate |

| Y Coordinates of field plots[1] | Y coordinates in SWEREFF 99 TM | NFI | Y Coordinate |
|---|---|---|---|
| National land use data | 10 m x 10 m | Swedish Environmental Protection Agency | CORINE |
| Soil moisture index | 10 m x 10 m | Swedish Environmental Protection Agency | SMI |

**Question:** How were annual and seasonal runoff quantified, and what was the resolution and quality of these data?

*Response*: *The annual and seasonal runoff were estimated using Hydrological Predictions for the Environment (HYPE) model, specifically S-HYPE which is the modified HYPE model for Swedish condition. S-HYPE is particularly suited for Sweden since there is a considerable variability in runoff conditions across different regions of Sweden and across different seasons. Here, S-HYPE was used to model seasonal and annual runoff in 33 605 sub-catchments across Sweden between 1982 and 2015. The modelled runoff was then designated as "Spring", "Summer", "Autumn", "Winter", and "Average". More details on this analysis was reported in our paper (Ågren et al., 2021).*

**Question:** Runoff data can surely have been at no finer scale than that of catchment level, in most cases. If so, how can it assist in mapping peat at 2 m resolution?

*Response*: *Yes the runoff data is modelled on a Sub-catchment scale (33 605 sub-catchments across the country, see map below under question 6). During the development of the SLU soil moisture map we included data capturing the controls on soil moisture in multiple scales, from regional weather patterns, down to local topography controlling how water is routed locally through the landscape. Our XGBoost model was trained using ca 16 000 field observations across the country and was built using 73 000 multiband raster stacks where each raster stack had a footprint of 2.5 km2 (i.e. the size of the original laser tiles). By resampling all 45 input-grids to 2 m grids to match the LIDAR-derived variables, we could allow the machine learning model to learn at what scale the major control on soil moisture appeared. While the runoff data (seasonal or annual) could be uniform within a tile it can help explain the variability between tiles in the model (i.e. large scale controls on soil moisture). Depth-to-water maps on the other hand showed a higher variability within a tile, but was in general "more uniform" between tiles across the country. It helped explain within tile variability in the model (i.e. local scale variability). In the feature reduction procedure variables that did not contribute significantly in predicting the soil moisture was removed from the model. Out of the evaluated 45 features, 28 were included in the final predictive model used to produce the SLU soil moisture map.*

*We showed that soil moisture is controlled on many scales. The terrain indices Depth-to-water and Topographic wetness index was the most important features for predicting soil moisture (Fig. 4A from Ågren et al., 2021) ("more within-tile local scale variability"). When it came to the seasonal weather patterns, autumn, summer and winter runoff contributed significantly and was included in the final predictive model (Fig. 4A from Ågren et al., 2021) ("more larger scale weather pattern, between tile variability").*

[Figure]

*Fig. 4A from Ågren et al 2021. Variable importance of the 28 input features for the 2-class (A) XGBoost model. The variable names are explained in Table 2. Note that the variable Coarse sediment was removed from the graph, as it was so close to 0 that the column became invisible.*

**Question:** In turn, more commentary was needed on the resolution and quality of the soil moisture maps.

***Response****: In the revised manuscript, we will add more information on how the model was trained and tested, to give more background on the quality measures (kappa and MCC).*

**Question:** What, for instance, is the extent of seasonal variability?

***Response:*** *See Figure 2 from (Lidberg et al., 2020) where (B) illustrates average winter runoff from the last +30 years and (C) average spring runoff from the last +30 years. Based on runoff modelling on >33000 subcatchments.*

[Figure]

*Figure 2 from Lidberg et al., 2020. An example of the variability of the landscape and climate in Sweden that could affect the hydrological modelling ("Other factors affecting the hydrological modelling"). Here exemplified by (A) the Swedish national DEM, (B) average winter runoff from the last 30 years and (C) average spring runoff from the last 30 years.*

**Question:** Is the parameter calculated perhaps an annual mean or median value?

***Response****: Runoff was modeled using S-Hype on a daily timestep, then summarized annually/seasonally, then averaged over the +30 year period.*

**Question:** The authors need to explain much more thoroughly the data used and the methods used in the machine learning model.

***Response****: As asked in the previous questions as well, we decided to expand the method section slightly to describe the production of soil moisture, however, for more detailed method description, we refer the readers to our recent papers; i.e. (Ågren et al., 2021; Lidberg et al., 2020). We now suggest to include the section below in the revised manuscript.*

***"2.3 Generating categorical (peat vs. mineral soils) and continuous organic layer thickness maps***

*This study utilized the SLU (Swedish University of Agricultural Science) soil moisture map that exhibits soil moisture variation across Sweden on an arbitrary scale from 1 to 100 i.e. from dry to wet (Ågren et al., 2021). The detailed methodology for producing the SLU soil moisture map was reported in (Ågren et al., 2021) and a previos verison of the map in (Lidberg et al., 2020). Here we give a brief introduction to the SLU soil moisture map. It was developed using a combination of digital terrain indices (derived from 2 m resolution digital elevation model based on airborne laser scanning data*

*(ALS) and ancillary data on quaternary deposits, soil depth, annual and seasonal runoff. The topographical indices were calculated on window sizes from 6 × 6 m to 160 × 160 m to allow for both large scale and small scale controls on soil moisture. By working on a higher resolution than most other studies, we aimed to improve the modelling of soil moisture in local pits and better capture the small-scale variability in riparian zones. In total, 45 different predictors (aka features) were evaluated for predicting soil moisture and after the feature reduction process, 28 predictors were remained in the final model. The predictors were utilized in an Extreme Gradient Boosting model (Chen et al., 2020) to predict the soil moisture across Sweden. The top predictors included Depth-To-Water index and Topographic Wetness Index calculated at different scales and resolutions, but also the autumn runoff and latitude (Ågren et al., 2021). The produced maps are now publicly available at www.slu.se/mfk. The model was trained and tested using 19 643 field observations from the NFI of which 80% were used for training and 20% was used for testing. The soil moisture map has Cohen's Kappa (Cohen, 1960) and Matthews Correlation Coefficient (MCC) (Matthews, 1975) values of 0.69 and 0.68, respectively. "*

**Question:** The predicted soil moisture data were then related to field-mapped peat depths collected from forestry surveys in which pits were excavated, and a regression model was fitted to the data. This is then used to predict peat thicknesses elsewhere across Sweden. However, the relationship between predicted organic layer thickness and measured thickness from the field survey data (Figure 4 in the ms.) shows enormous scatter. The bulk of the data points appear to be for quite thin organic layers (bottom left-hand corner of Fig 4), with relatively few observations > 60 cm (right hand part of Fig 4).

*Response: The aim of this study was to develop and evaluate a new method for delineating the horizontal distribution of peat. Despite some limitations of the proposed method, it predicts the majority of the observations correctly and provides an effective approach for predicting peat distribution at a landscape scale. Due to the overlap of the points in the figure, it was difficult to detect this pattern. Therefore, we now revised the figure to include the quadrants of the confusion matrix, using ≥50 cm peat as an example. If we take the example of delineating peat with ≥50 cm depth, it is true that the model misclassifies peat as mineral soil in 104 instances (FN) and misclassifies mineral soils as peat in 135 cases (FP) But, 427 observations are correctly classified as peat and 2217 are correctly classified as mineral soils. 2644 of 2883 soil pits are correctly classified in this example. This gives the ≥50 cm peat map a higher overall quality (Kappa and MCC) than the topographic and Quaternary deposits maps. These findings highlight that the prediction was substantially better than what was deemed in the previous version of Figure 4.*

[Figure]

*Figure 4 from the article with the quadrants for the confusion matrix highlighted in red, here using ≥50 cm peat as an example. Predicted vs measured continuous thickness of the organic layer in the evaluation dataset (n = 2 883). Dashed line indicate the 1:1 line and black line a linear regression ($R^2$ = 0.67, p<0.001), grey lines indicate 95% CI. Field measurements of peat thickness that were 99 cm or above were reported as 99 cm, hence the many over lapping data points.*

**Question:** The authors do not actually describe the process of producing their predicted organic layer maps from the soil moisture data, but rather simply jump from Fig 3 to a discussion of the resulting maps. This needs to be corrected.

***Response:*** *We address the comment by adding this paragraph in the manuscript: "By solving Eq. 7 for X when Y was 30, 40 and 50 cm we could determine the soil moisture limits for classifying peat soil. At organic layer thickness of ≥30, ≥40, and ≥50 cm, the soil moisture limits were ≥76%, ≥83% and ≥87%, respectively (Figure 3). These thresholds were used to reclassify the soil moisture map into maps of peat extent while the remaining soil was delineated as mineral soil. Hence, three different peatland maps were derived, which we referred as "peat ≥ 30 cm", "peat ≥ 40 cm", and "peat ≥50 cm". In addition, a continuous organic layer thickness map was generated by applying Eq. 7 in raster calculator on the continuous soil moisture map. This continuous map does not contain discrete classes of mineral and peat soils, rather, it presents the distribution of organic layer thickness across the landscape."*

**Question:** Given the enormous scatter in Fig 4, the authors at several places say that their thickness maps should not be 'taken literally' (e.g. line 460, line 466) and yet there is no real quantification of the probable magnitude of error at any location. This could have been done by comparing with the field data acquired from pits. The RMSE was reported as 19 cm (line 306) but this is a huge uncertainty given that most of the organic layers appear to be less than 20-30 cm in thickness. Is this level of uncertainty actually acceptable, and are the predicted depths sufficiently reliable for the estimation of carbon stocks, for instance?

*Response: We deem the level of uncertainty to be satisfactory for the horizontal delineation of peatlands; especially if the continuous map are used to highlight the areas where the delineation is more uncertain, i.e. along the borders of the peatlands. However, given the large RMSE for the depth estimates, and the generally thin layers of organic soils across most of the Swedish forest landscape, we do not suggest to use the depth estimates for carbon stocks. We will clarify this in the discussion.*

**Question:** Overall, I was left unsure about how much confidence could be placed in the thickness maps generated by the authors. I think that a fuller discussion of actual thicknesses and the likely uncertainty (surely varying with topographic position, and perhaps areal extent of particular organic or peat deposits) in the predictions is required.

*Response: Thank you for your comment, however, providing more details of the thickness map was not within the main scope of the study. Rather, we focused on the horizontal delineation of peat. We will add some text in the discussion to clarify this.*

**Question:** The authors claim excellent resolution in mapping peat deposits covering just 4 m$^2$ (e.g. line 405) - i.e., just a single pixel in data at 2 m resolution. Do such tiny peat deposits actually exist? If so, what accounts for their isolated accumulation? The authors need to comment.

*Response: While there exist really small topographic hollows (in the order of 4m2) that can fill up with peat, this is not the typical peat that we were able to map with the new methodology. The visual inspection of the map indicate that the main improvement from traditional maps is that the maps capture the riparian peat, and forested peatlands that were misinterpreted as mineral soil from aerial photos. It also gives a much more accurate delineation of the border between for example a flat mire and surrounding drumlins. This is easy to see based on ALS data while this was more difficult using aerial photos. We will now explain this further in the revised manuscript.*

**Question:** There are minor errors scattered throughout the ms. In particular, I would suggest that as a formal geological Period, 'Quaternary' should be capitalised. This is written 'quaternary' at many places in the ms., and all instances need correcting. The authors are occasionally inconsistent with this, such that Table 2 for instance contains 'Quaternary' as does the heading for Section 2.4, but elsewhere, mostly lower-case letters are used.

*Response: We will make sure to capitalize Quaternary throughout the manuscript. We will also ensure other necessary minor improvements.*

*Additional author comment: While we were revising our manuscript we found an error in Figure 5. The thickness of the organic layer should range 6-88 cm. Should this article be accepted for publication we will revise the figure and the text in the revised manuscript accordingly.*

**References**

Chen, T., He, T., Benesty, M., Khotilovich, V., Tang, Y., Cho, H., Chen, K., Mitchell, R., Cano, I., Zhou, T., Li, M., Xie, J., Lin, M., Geng, Y., and Li, Y.: xgboost: Extreme Gradient Boosting. R package version 1.0.0.2. [code], 2020.

Cohen, J.: A Coefficient of Agreement for Nominal Scales, Educational and Psychologial Measurment, 20, 37-46, https://doi.org/10.1177/001316446002000104, 1960.

Lidberg, W., Nilsson, M., and Agren, A.: Using machine learning to generate high-resolution wet area maps for planning forest management: A study in a boreal forest landscape, Ambio, 49, 475-486, 2020.

Matthews, B. W.: Comparison of the predicted and observed secondary structure of T4 phage lysozyme, Biochimica et Biophysica Acta (BBA) - Protein Structure, 405, 442-451, doi:10.1016/0005-2795(75)90109-9, 1975.

Ågren, A. M., Larson, J., Paul, S. S., Laudon, H., and Lidberg, W.: Use of multiple LIDAR-derived digital terrain indices and machine learning for high-resolution national-scale soil moisture mapping of the Swedish forest landscape, Geoderma, 404, 115280, https://doi.org/10.1016/j.geoderma.2021.115280, 2021.

 **Citation**: https://doi.org/10.5194/egusphere-2022-79-EC1

---

## Author Comment (AC2)

**Answer to questions by RC1: Anonymous Referee #1**

**RC1: Anonymous Referee #1, 16 Aug 2022**

Ågren et al. map the spatial distribution of peat soils and organic layer thickness in Sweden using an existing soil moisture map and national-level field inventory data. The manuscript is well written and mostly sound, but I was left partly confused when reading the manuscript. I have the following major points.

**Question**: Why did you predict thickness of organic layer based on soil moisture map and not from the original predictor variables that were used to produce the soil moisture map? This seems to be quite odd as there is now double uncertainty in the estimates, as the prediction of soil moisture was already a little uncertain. You should justify your approach better. It would be also interesting to compare different predictor variable sets (e.g. topography variables, satellite imagery etc.) to produce the thickness of organic layer and not just use one existing map.

*Response: There is an urgent need for an improved peat map among the Swedish land managers and practitioners.  After the SLU soil moisture map was recently produced at a remarkably high resolution, there has been wide interest how we can understand the relationship between the soil moisture variation and distribution of peat across the country using the SLU soil moisture map. This present study was an immediate response to this applied research need. We understand your concerns regarding uncertainty, however, our comparison with "traditional maps" suggest that the peat delineation based on our approach was satisfactory (Table 3 in our article).*

**Question**: Based on Figure 4, the fit of the model between soil moisture and organic layer thickness is not very good. This should be discussed in more detail.

*Response: Unfortunately, the fit of the model in figure 4 looks worse than it actually is due to the overlap of the points in the figure. We have therefore revised the figure to include the quadrants of the confusion matrix, using ≥50 cm peat as an example. If we take the example of delineating peat with ≥50 cm depth, the model misclassifies peat as mineral soil in 104 instances (FN) and misclassifies mineral soils as peat in 135 cases (FP) But, 427 observations are correctly classified as peat and 2217 are correctly classified as mineral soils. 2644 of 2883 soil pits are correctly classified in this example. This gives the ≥50 cm peat map a higher overall quality (Kappa and MCC) than the topographic and Quaternary deposits maps. We will include this new graph in the revised manuscript and explain this in the discussion.*

[Figure]

*Figure 4 from the article with the quadrants for the confusion matrix highlighted in red, here using ≥50 cm peat as an example. Predicted vs measured continuous thickness of the organic layer in the evaluation dataset (n = 2 883). Dashed line indicate the 1:1 line and black line a linear regression ($R^2$ = 0.67, p<0.001), grey lines indicate 95% CI. Field measurements of peat thickness that were 99 cm or above were reported as 99 cm, hence the many over lapping data points.*

**Question**: Actually, there has been a lot of discussion that R2 should not be used for nonlinear regressions. You should justify why you evaluated model performance with R2 and you should also report how you calculated R2.

*Response: In our selection of model (Figure 3) we first disregarded the non-significant models (based on P-values). Then we used R2 to rank them because that provided the best accuracy metrics for the curve estimation procedure in SPSS we used. In regression analysis, the more parameters you add the more you increase the R2, however, you risk over-parametization of the model.  Normally, we would never use a cubic model for predictions as it often overfits the model and introduce bias. However, in this case it was the only function that could capture the variability of the data with a sharp increase at the low end of soil moisture, followed by a plateau and then a sharp increase on the high end. However, it could not fully capture this pattern, hence field measurements ranged 0-99 cm while predictions ranged 6-88 cm. If our focus had been on predicting the thickness of the organic layer then we would have investigated a more complex non-linear model that might have captured this better. However, in this study the thickness of the organic layer is just a means to an end, as we focus on the horizontal delineation of peat. The red curve in the graph fits the data "well" in the range*

*we focus on (ca 70-90% soil moisture), in the green color-range.*

*The sharp increase on the low end is explained by the fact that the driest sites are generally found on crests and ridges characterized by rock outcrops with very thin organic layers. The plateau in the middle represents typical forest soils (mostly podsoils) and the sharp increase at the high end is explained by the formation of peat in the wettest areas. So in this case we believe it is an appropriate use of the cubic function.*

**Question**: Precision was lower in your map than in the topographic map. It seems that your mapping approach seems to overpredict peatlands, at least to some extent. This seems to be the case also when visually interpreting the material in Figs. 5 and A1 and looking at the information in Table 3.

This should be accounted for and discussed in more detail. You could discuss e.g. why your approach seems to overpredict the extent of peatlands and overestimate thickness of thin organic layers. The overestimation of thickness of thin organic layers is probably due to the selected cubic model. You could potentially also use other (non-linear) regression models and discuss the pros and cons of different models.

| | TP | TN | FP | FN | Accuracy (%) | Precision (%) | Recall (%) | Specificity (%) | Kappa | MCC |
|---|---|---|---|---|---|---|---|---|---|---|
| *Peat ≥50 cm map* | 427 | 2217 | 135 | 104 | 91.7 | 75.9 | 80.4 | 94.3 | 0.73 | 0.73 |
| *Peat ≥40 cm map* | 417 | 2133 | 165 | 114 | 90.1 | 71.7 | 78.5 | 92.8 | 0.69 | 0.69 |
| *Peat ≥30 cm map* | 537 | 2013 | 199 | 134 | 88.5 | 72.9 | 80.0 | 91.1 | 0.69 | 0.69 |
| Topographic map | 264 | 2318 | 39 | 266 | 89.4 | 87.1 | 49.8 | 98.4 | 0.58 | 0.61 |

| | | | | | | | | | | |
|---|---|---|---|---|---|---|---|---|---|---|
| Quaternary deposits map | 363 | 2227 | 130 | 167 | 89.7 | 73.6 | 68.5 | 94.5 | 0.65 | 0.65 |

**Response:** *Mostly it is the topographic map that under-predicts peatlands. For example, the topographic map only over-predicts 39 instances but under-predicts 266 instances, a clear bias toward under-prediction (with an overweight of 227 misclassified soil pits). So the high precision for the topographic map is driven by the low number of FP, however, note that the recall for the topographic map was below 50%. Our ≥50 cm peat under-predicts peat in 104 instances and over-predicts peat in 135 instances, a fairly balanced distribution of errors with only a slight overweight (n=31) towards over-predictions. But you are correct, that this problem is bigger for the peat ≥30 cm map, with an overweight of 65 plots towards overprediction. This illustrates how easy it is to misinterpret results when using measures that only focus on parts of the confusion matrix. The best measure of the overall performance of the map quality (taking into account both over- and under-predictions) is the MCC which shows that our predicted maps outperform the topographical and Quaternary deposits maps and that the ≥50 cm peat has the highest quality. We suggest to clarify this further in the discussion in the revised manuscript and we will also add text on the slight over-prediction of peat on the ≥30 cm maps.*

*As for the visual interpretation of the map, we were also guilty of this interpretation at first because we are so used to believe that the traditional maps show the truth. However, the evaluation of the map in Table 3 shows that our predicted maps locate more peatlands (higher recall rate), without over-predictions of peatlands (for peat ≥50 cm). It took us some time to "recalibrate" our brain and not trust the classical topographical maps "blindly" when interpreting the maps. And that we are considering as a success of our proposed methodology for peatland delineation.*

Additionally, I have the following more detailed but mostly minor comments:

Abstract:

**Comment:** Please remove the first sentence. It is not necessary.

**Response**: *The first sentence will be deleted.*

**Comment:** l17-19: it is not needed to report the results from an existing study. Please rephrase and shorten the sentence

**Response:** *"with high accuracy (Kappa = 0.69, MCC = 0.68)" will be removed from the sentence.*

**Comment:** l25: please report also the precision results

**Response:** *We choose not to focus on the precision, but we will motivate better in the discussion why.*

**Comment:** l28: "peatlands visible from airplanes" could be written "peatlands that can be visually detected from aerial imagery"

*Response: We will follow this suggestion*

**Comment:** l29: delete "most importantly"

*Response: "most importantly" will be deleted.*

Introduction:

**Comment:** l80-87: the direct quote is unnecessarily long. Do you need to include it?

*Response: We believe it's important to be clear about this definitions and would like to keep it as it reads.*

**Comment:** l113: is the fourth objective necessary to include?

*Response: Yes, we believe so. According to "Digital mapping of peatlands - A critical review" by (Minasny et al., 2019) the review showed that while it is common to delineate peat extent, studies rarely perform validation or calculations of the uncertainty of the predictions. We believe it is a good practice in research to provide uncertainty estimates of model results.*

**Comment:** l114: write "study provides a guide to map…"

*Response: We followed this suggestion*

Methods:

**Comment:** l142-155: this paragraph could be shortened as it describes results from an earlier study, not the methods of this study.

*Response: While you asked us to shorten this section, the editor asked us to expand this section. We decided to add some more background on the soil moisture map which probably will aid the reader to better understand this article. Please see our earlier response to the editor.*

**Comment:** Did you account for spatial autocorrelation when e.g., constructing the model and dividing the calibration and validation datasets?

*Response: No, as the closest distance between soil pits in the datasets are 500 m and autocorrelation in peat depth is usually detected on a smaller scale, we could not account for the autocorrelation. Hence the data was just split randomly into two datasets. Although this may be interesting to investigate further in the future, we argue that autocorrelation may not produce better results because of the sample design of the field dataset.*

**Comment:** l195: This is difficult to understand. Does it mean that 1:25 000 map covers 1.7% of Sweden and so on?

*Response: Yes, we now propose to remove the brackets in hope that this makes it clearer.*

*"However, there are various scales with different coverages for the quaternary deposit maps in Sweden, such as 1:25 000 covers 1.7% of the area, 1:50 000 covers 2.7%, 1:100 000 covers 47%, 1:200 000 covers 1.4%, 1:250 000 covers 21.2%, 1:750 000 covers 33.6% and 1:1 000 000 covers 100%."*

**Comment:** Why did you include the used accuracy metrics? Kappa has been heavily criticized (see e.g., https://doi.org/10.1016/j.rse.2019.111630). You could also have included F-score.

***Response****: Yes, Kappa has been heavily criticized, but since it has been used extensively previously and is still often used in map-quality studies we present it here for easy comparison with other studies. However, due to the criticism towards Kappa we also present MCC which is considered a better measure (Chicco et al., 2021).*

*We choose MCC instead of F-score as MCC has been shown to have advantages over the F-score (Chicco and Jurman, 2020). However, as we publish the raw confusion matrix with this manuscript, it is possible for the interested readers to get an estimate F-score (along with other metrics of choice) if deemed necessary. We believe it should be a standard to publish the raw confusion data (TP, TN, FP, FN) as that will enable future meta-studies to calculate all possible evaluation metrics needed. This is unfortunately quite often neglected in the literature today.*

*We will also add a sentence in the discussion to explain this further; "Out of the two measurements kappa and MCC, MCC is considered the most informative measure (Chicco et al., 2021)."*

**Comment:** Section 2.6: How were the field inventory datasets upscaled? Does this simply mean that you calculated national level statistics from the datasets using different methods?

***Response****: Yes, it is simple statistical upscaling performed by the experts in charge of the national surveys. They are expert statisticians and have a deep understanding of the sampling design and how to best calculate national estimates based on the survey data. In the revised manuscript, we will now explain that we use statistical upscaling and that is was performed by the experts.*

*"To compare the predicted peat soil estimates from the maps with other estimates of peatland coverage in Sweden, we also calculated peatland coverage by **statistical** upscaling from the national inventories, i.e. NFI and SFSI (SLU, 2021) to derive a complete coverage for the Swedish forest landscape. From these inventories, peat coverage was estimated by the **statistical experts** at NFI **(Fridman et al., 2014)** and SFSI **(Stendahl et al., 2017)** in 6 different ways".*

*We reference Nilsson et al., 2018 and Hånell, 2009 where the upslacling calulations are described more-in depth. We now also refer to the articles by (Fridman et al., 2014), and (Stendahl et al., 2017) that describes the NFI and SFSI in more detail.*

**Comment:** the heading of 3.4 could be changed. Should it be "visual interpretation of peatland maps"?

***Response:*** *Good suggestion, we will change it accordingly.*

Discussion and conclusions

**Comment:** l375-376: This is misleading as you used ALS data very indirectly.

*Response*: *We kindly disagree here, the whole foundation of the soil moisture map is ALS data. Out of the 28 input data, the most important variables for predicting soil moisture were the digital terrain indices derived from ALS-data, not the traditional maps or the runoff. We believe this to be clearer now in the revised manuscript that we have expanded the methods section on the SLU soil moisture map.*

**Comment:** l460: you write multiple times that the map should not be taken literally. It is not necessary to mention this multiple times.

*Response*: *We will address this throughout the manuscript.*

**Comment:** The section "The novelty of the developed maps" could be shortened and merged with conclusion section. Some text can also be moved to other parts of discussion.

*Response*: *We will consider this in our revised manuscript.*

**Comment:** l509: delete "coarse", "global mapping" is sufficient.

*Response*: *We will follow this suggestion*

**Comment:** l504-510: Sentinel-2 has 10 m resolution and it surely can be used for quite detailed planning. There is also other remote sensing than just ALS data that can be used in detailed planning.

*Response:* *Yes, sentinel-2 data can be used to detect open peatlands, however, it shares the same limitation as aerial photos that the canopy cover makes it difficult to detect the soils in large areas of the Swedish forest landscape. A benefit of our method of mapping peat using mostly ALS data is that the ALS "sees through the canopy cover". For more detailed planning, for example designing a riparian zone (which in Sweden is on average 4 m wide), or finding suitable crossings across streams, high resolution aerial photos (often 0,5 m resolution) in combination with ALS data is usually preferred by practitioners. Sentinel-2 data may be too "pixelated" for such planning. When we developed the soil moisture map (Ågren et al., 2021), we evaluated the CORINE land use data which is based on Sentinel-2. However it was excluded due to low contribution to the models. See also our answer to editor's questions.*

**References**

Chicco, D. and Jurman, G.: The advantages of the Matthews correlation coefficient (MCC) over F1 score and accuracy in binary classification evaluation, Bmc Genomics, 21, 2020.
Chicco, D., Warrens, M. J., and Jurman, G.: The Matthews Correlation Coefficient (MCC) is More Informative Than Cohen's Kappa and Brier Score in Binary Classification Assessment, Ieee Access, 9, 78368-78381, 10.1109/Access.2021.3084050, 2021.
Fridman, J., Holm, S., Nilsson, M., Nilsson, P., Ringvall, A. H., and Stahl, G.: Adapting National Forest Inventories to changing requirements - the case of the Swedish National Forest Inventory at the turn of the 20th century, Silva Fenn, 48, 2014.
Minasny, B., Berglund, O., Connolly, J., Hedley, C., de Vries, F., Gimona, A., Kempen, B., Kidd, D., Lilja, H., Malone, B., McBratney, A., Roudier, P., O'Rourke, S., Rudiyanto, Padarian, J., Poggio, L., ten Caten, A., Thompson, D., Tuve, C., and Widyatmanti, W.: Digital mapping of peatlands - A critical review, Earth-Sci Rev, 196, 2019.

Stendahl, J., Berg, B., and Lindahl, B. D.: Manganese availability is negatively associated with carbon storage in northern coniferous forest humus layers, Sci Rep-Uk, 7, 2017.
Ågren, A. M., Larson, J., Paul, S. S., Laudon, H., and Lidberg, W.: Use of multiple LIDAR-derived digital terrain indices and machine learning for high-resolution national-scale soil moisture mapping of the Swedish forest landscape, Geoderma, 404, 115280, https://doi.org/10.1016/j.geoderma.2021.115280, 2021.

**Citation**: https://doi.org/10.5194/egusphere-2022-79-RC1

---

## Author Comment (AC3)

**Answer to questions by RC2: Anonymous Referee #2**

**RC2: Anonymous Referee #2, 23 Aug 2022**

**Comment:** In this paper the authors present an approach for the identification of peat soils by fitting an empirical relationship between the thickness of the organic layer - measured across Sweden - and a continuous soil moisture map. The article is very interesting and quite well written, but I would prefer to see a bit more detailed description of the procedure followed by the authors.

*Response: We will give more details in our description of the procedure. We also highlighted this addition in previous answers to the editors and reviewers.*

English language and style are quite fine, only a minor spell check is required.

Most of the references are listed without specifying the number of pages. Please carefully check them, since other mistakes are present. All the references in other languages than English should be clearly indicated.

Other specific comments:

**Comment:** Line 54: Sewell et al (2020) is not reported in the reference list.

*Response: (Sewell et al., 2020) will be added to the reference list.*

**Comment:** Line 142: please define the acronym SLU, since the readers could not be familiar with it.

*Response: We will add the English explanation to SLU (Swedish university of Agricultural Science)*

**Comment:** Line 148: please add references for these statistics.

*Response: We will add (Cohen, 1960) and (Matthews, 1975).*

**Comment:** Line 387: Arrouays et al (2014) is not reported in the reference list.

*Response: (Arrouays et al., 2014) will be added to the reference list.*

**Comment:** Line 388: Jackson et al (2017) is not reported in the reference list.

*Response: (Jackson et al., 2017) will be added to the reference list.*

**Comment:** Line 458: Nijp et al (2019) is not reported in the reference list.

*Response: (Nijp et al., 2019) will be added to the reference list.*

I recommend to revise the article. Minor revision is necessary.

**References**
Arrouays, D., Grundy, M. G., Hartemink, A. E., Hempel, J. W., Heuvelink, G. B. M., Hong, S. Y., Lagacherie, P., Lelyk, G., McBratney, A. B., McKenzie, N. J., Mendonca-Santos, M. d. L., Minasny, B., Montanarella, L., Odeh, I. O. A., Sanchez, P. A., Thompson, J. A., and Zhang, G.-L.: Chapter Three - GlobalSoilMap: Toward a Fine-Resolution Global Grid of Soil Properties, in: Advances in Agronomy,

edited by: Sparks, D. L., Academic Press, 93-134, https://doi.org/10.1016/B978-0-12-800137-0.00003-0, 2014.

Cohen, J.: A Coefficient of Agreement for Nominal Scales, Educational and Psychologial Measurment, 20, 37-46, https://doi.org/10.1177/001316446002000104, 1960.

Jackson, R. B., Lajtha, K., Crow, S. E., Hugelius, G., Kramer, M. G., and Pineiro, G.: The Ecology of Soil Carbon: Pools, Vulnerabilities, and Biotic and Abiotic Controls, Annu Rev Ecol Evol S, 48, 419-445, 10.1146/annurev-ecolsys-112414-054234, 2017.

Matthews, B. W.: Comparison of the predicted and observed secondary structure of T4 phage lysozyme, Biochimica et Biophysica Acta (BBA) - Protein Structure, 405, 442-451, doi:10.1016/0005-2795(75)90109-9, 1975.

Nijp, J. J., Metselaar, K., Limpens, J., Bartholomeus, H. M., Nilsson, M. B., Berendse, F., and van der Zee, S. E. A. T. M.: High-resolution peat volume change in a northern peatland: Spatial variability, main drivers, and impact on ecohydrology, Ecohydrology, 12, 2019.

Sewell, P. D., Quideau, S. A., Dyck, M., and Macdonald, E.: Long-term effects of harvest on boreal forest soils in relation to a remote sensing-based soil moisture index, Forest Ecol Manag, 462, 117986, https://doi.org/10.1016/j.foreco.2020.117986, 2020.

**Citation**: https://doi.org/10.5194/egusphere-2022-79-RC2

---

## Author Response (AR1)

**Letter to Editor regarding minor revisions of EGUSPHERE-2022-79 "Delineating the distribution of mineral and peat soils at the landscape scale in northern boreal regions" by Anneli M. Ågren, Eliza Maher Hasselquist, Johan Stendahl, Mats B. Nilsson, and Siddhartho S. Paul.**

As for a point-by-point reply to the comments, se our earlier answer to the editor and reviewer questions where we addressed all points.

https://doi.org/10.5194/egusphere-2022-79-AC1
https://doi.org/10.5194/egusphere-2022-79-AC2
https://doi.org/10.5194/egusphere-2022-79-AC3

In this revised manuscript with tracked changes we have addressed all comments according to our answers in the above documents except in one instance: Referee 2 gave the following comment:

Comment: The section "The novelty of the developed maps" could be and merged with conclusion section. Some text can also be moved to other parts of discussion.

To which we answered:  We will consider this in our revised manuscript.

Our new response is: We have seriously considered the suggestion by referee #2. After thoughtful digestion of the two sections we prefer to keep them as is. The two sections addresses two different focuses. The section **"The novelty of the developed maps"** addresses the relation between this study and the recent progress internationally attempting to improve mapping of peatlands. The section **"Conclusions"** detail out what peat cover characteristics that can be described by the approach described in this article, not possible by earlier approaches. Thus the two section have similar, but still clearly different, focuses and thus we prefer to keep them as is.

In addition to addressing the comments we also:

- Clarified the calculations of peat in Table 3. In the methods we now write "6) peat with no thickness restriction (i.e. peat if organic layer ≥ 30 cm + peaty mor with organic layer < 30 cm)." and changed Table 3 to "6. Peat coverage with no peat thickness restriction according to upscaling from SFSI".

- We also revised the reference list and added article no, doi's, etc.

- We provide 2 new figures (Fig04 and Fig 05)

We hope the manuscript can now be accepted for publication.

Sincerely, Anneli Ågren and co-authors.